# Effect of Ginseng Sapogenin Protopanaxadiol-Enriched Rice (DJ-PPD) on Immunomodulation

**DOI:** 10.3390/plants12040767

**Published:** 2023-02-08

**Authors:** Chaiwat Monmai, Jin-Suk Kim, So-Hyeon Baek

**Affiliations:** Department of Agricultural Life Science, Sunchon National University, Suncheon 59722, Republic of Korea

**Keywords:** transgenic rice, protopanaxadiol, ginseng root extract, anti-inflammation, immune enhancement, immunomodulation, NF-κB, MAPK

## Abstract

Protopanaxadiol (PPD), a gut microbiome-induced ginseng metabolite, has positive immune effects. We previously reported the immune-boosting and anti-inflammatory effects of PPD-enricshed rice seed extracts in normal and inflammatory cell environments, respectively. In the present study, the immunomodulatory activity of PPD-enriched transgenic rice seed extract (DJ-PPD), which exhibited the highest immune-related activity among all available extracts, was compared with that of commercially synthesized 20s-PPD (S-PPD) and natural ginseng root extract (GE), in RAW264.7 cells. Compared with S-PPD and GE treatment, DJ-PPD treatment (i) significantly promoted NF-κB p65 and c-Jun N-terminal protein kinase (JNK) phosphorylation; (ii) upregulated *IL-1β*, *IL-6*, *COX-2*, *TLR-4*, and *TNF-α* expression; (iii) and increased prostaglandin E_2_ (PGE_2_) production. However, there were no significant differences in the effects of the three treatments containing PPD-type sapogenin or saponins on nitric oxide (NO) production and phagocytic activity. In the inflammatory cell environment, DJ-PPD treatment markedly decreased the production of LPS-induced inflammatory factors, including NO and PGE_2_, as well as proinflammatory cytokine expression, by decreasing phosphorylated (p-)NF-κB p65, p-p38 MAPK, and p-JNK levels. Thus, DJ-PPD that does not require complex intestinal microbial processes to exert higher anti-inflammatory effects compared with S-PPD and GE. However, DJ-PPD exerted similar or higher immune-boosting effects (depending on inflammatory biomarkers) than S-PPD and GE. These findings indicate the potential of PPD-enriched transgenic rice as an alternative immunomodulatory agent.

## 1. Introduction

Functional foods, also called nutraceuticals, are either whole foods or food ingredients associated with major health benefits [1], such as antioxidant, anti-inflammatory, anticancer, and immune-boosting effects, all of which are provided by their biologically active compounds [1,2,3]. The increasing prevalence of various diseases has sparked interest in nutraceuticals [4].

Protopanaxadiol (PPD; Appendix A), one of the main active compounds found in ginseng (*Panax ginseng* Meyer), has repeatedly shown pharmaceutical potential. For example, Dong et al. [5] assessed the cytotoxicity of a ginseng extract (GE) and PPD-type ginsenosides against various types of cancer cells (e.g., lung, colon, and breast cancer, as well as human liver carcinoma cells), finding that PPD exerts cytotoxic effects on these cells. In addition, Jin et al. [6] and Lee et al. [7] reported the antitumoral activities of PPD on breast carcinoma cells. Furthermore, in an immunosuppressive model, PPD-type ginsenosides exhibited immune-boosting effects by inducing CD4+ and CD8+ lymphocyte proliferation [8]. Kim and Cho [9] suggested that 20S-dihydroprotopanaxadiol enhances phagocytic uptake in macrophages. Moreover, Wang et al. [10] evaluated the effects of PPD on colorectal cancer. They reported that PPD treatment and the combination of PPD with chemotherapeutic agent fluorouracil (5-FU) resulted in vitro and in vivo anticancer activity. PPD significantly suppressed HCT-116 cell proliferation in a dose-dependent manner (up to 35 µM). Treatment with 20 µM of PPD increased apoptosis in HCT-116 cells to 9.1 ± 0.7% compared to the untreated group (5.0 ± 0.6%). These in vitro data were confirmed using a xenografted athymic mouse model, and PPD (15 and 30 mg/kg body weight) significantly (*p* < 0.5) reduced the tumor weight. Aside from its the anticancer and immune-boosting effects, PPD also exerts anti-inflammatory effects by suppressing the regulation of lipopolysaccharide (LPS)-activated p38 and c-Jun N-terminal protein kinase (JNK) pathways [11]. Similarly, Lee et al. [12] found that the anti-inflammatory effects of PPD were due to the suppression of nitric oxide (NO) and prostaglandin E_2_ (PGE_2_) production, as well as proinflammatory cytokine expression. In addition, other antistress [13], antioxidant [14] and central and peripheral nervous system-related [15] effects of PPD have been reported.

PPD-producing transgenic rice was generated by introducing *P. ginseng* dammarenediol-II synthase and protopanaxadiol synthase genes to Dongjin rice in our laboratory [16]. The expression of the introduced genes was confirmed in the first and second generation of these transgenic lines using genomic PCR and qPCR, respectively. Then, the fourth generation of this transgenic rice grain was randomly selected for evaluation of the immunomodulatory activity in RAW264.7 and LPS-stimulated RAW264.7 cells [17]. We found that transgenic rice line 8 seed extract (DJ-PPD), which contained the highest amount of PPD (7.28 ± 0.64 µg of PPD/g of rice seed extract) (Appendix A), exhibited the strongest immunomodulatory activity among all tested transgenic rice lines. Indeed, DJ-PPD enhanced NO and PGE_2_ production, immune-related mRNA expression levels, and phagocytic activity by activating the NF-κB and MAPK pathways in RAW264.7 cells. Moreover, treatment with DJ-PPD significantly inhibited LPS-induced inflammation factors. In addition, certain anti-inflammatory and immune-boosting activities were observed in the DJ (normal rice without PPD) group. Additionally, DJ-PPD markedly suppressed oxidative and melanogenic activities in melan-a cells, with the highest power among all transgenic rice lines [18]. The oxidative activities of transgenic rice seed extracts were correlated with their PPD contents. Based on these results, we hypothesized the existence of cooperative effects between the transgenic PPD and original compounds in rice. Therefore, we further investigated DJ-PPD, which exhibited the highest immunomodulatory, antioxidative, and melanogenic activities among all the transgenic rice lines in the present study. Specifically, the aim of the present study was to evaluate the immune-related effects of DJ-PPD, i.e., its immuno-boosting effect in RAW264.7 cells and anti-inflammatory effects in RAW264.7 cells under LPS-induced inflammation, in comparison to the effects of commercially synthesized PPD (S-PPD) and natural ginseng root (GE).

## 2. Results

### 2.1. Cell Viability and NO Production in LPS-Stimulated RAW264.7 Cells

Prior to the in vitro immunomodulatory effects assay, the cytotoxicity of all treatments was evaluated on LPS-stimulated RAW264.7 cells. After 24 h of LPS stimulation, the treated cells were used to evaluate cell viability. Under the dose range used in this study, treatment with DJ (normal rice without PPD), DJ-PPD, S-PPD, GE, aspirin (positive control), or dimethyl sulfoxide (DMSO), together with 1 µg/mL of LPS, did not exert toxic effects on RAW264.7 cells (Figure 1). In addition, treatment with DJ (up to 100 µg/mL), DJ-PPD (up to 100 µg/mL), GE (up to 50 µg/mL), and S-PPD (up to 700 pg/mL) enhanced proliferation in LPS-induced RAW264.7 cells.

An LPS model served as a proxy for an inflammation environment. NO was detected in LPS-treated cells but not in Roswell Park Memorial Institute (RPMI)-treated cells (Table 1). Treatment with DJ, DJ-PPD, S-PPD, and GE significantly reduced LPS-induced NO production in a concentration-dependent manner. Compared with the DJ treatment, treatment with DJ-PPD, S-PPD, and GE at low concentrations (10 µg/mL or 70 pg/mL) significantly decreased NO production. In addition, treatment with DJ-PPD at high concentrations (50 and 100 µg/mL) caused the highest inhibition of LPS-induced NO production among all tested treatments. The results show that the LPS-induced NO production was reduced depending on the concentration of the samples without any cytotoxicity on cells (up to 100 µg/mL (DJ, DJ-PPD, and GE) or 700 pg/mL (S-PPD)).

### 2.2. Cellular Viability and NO Production in RAW264.7 Cells

RPMI and 0.1% DMSO treatments had similar effects on cell viability and NO production (Figure 2 and Table 2), indicating that 0.1% DMSO does not exert cytotoxicity or affect NO production in RAW264.7 cells. Compared with RPMI treatment, treatment with DJ, DJ-PPD, S-PPD, and GE (up to 100 µg/mL or 700 pg/mL) exerted similar or better effects on cell viability.

Compared with DJ (normal rice without PPD)-treated cells, NO production was significantly enhanced in cells treated with DJ-PPD, S-PPD, and GE; furthermore, NO production increased in a treatment-concentration-dependent manner. However, at high concentrations (50 and 100 µg/mL or 350 and 700 pg/mL), there were no significant differences in the level of NO production among DJ-PPD-, S-PPD-, and GE-treated cells. This indicates the immune-enhancing activity of PPD-type sapogenin or saponins, which induce the production of NO in a concentration-dependent manner.

### 2.3. Immune-Related mRNA Expression in LPS-Stimulated RAW264.7 Cells

To investigate the anti-inflammatory activities of DJ-PPD, the LPS-induced proinflammatory genes were evaluated. Various immune-related genes (*IL-1β*, *IL-6*, *COX-2*, *iNOS*, *TLR-4*, and *TNF-α*) were upregulated in LPS-treated cells (Figure 3). However, LPS-induced proinflammatory gene expression was downregulated in cells treated with DJ, DJ-PPD, S-PPD, and GE. Aspirin treatment (the positive control) produced the highest inhibition of proinflammatory gene expression. Among all treatments, pretreatment with DJ-PPD at 100 µg/mL significantly suppressed the expression of LPS-induced proinflammatory mRNAs; this suppression level was higher than that caused by the other treatments. However, there were no significant differences in the mRNA expression levels of S-PPD- and GE-treated, LPS-stimulated RAW264.7 cells. This result suggests that PPD-enriched rice DJ-PPD has an anti-inflammatory effect because it suppressed the production of LPS-induced proinflammatory genes.

### 2.4. Immune-Related mRNA Expression in RAW264.7 Cells

As shown in Figure 4, the expression levels of *IL-1β*, *IL-6*, *COX-2*, *iNOS*, *TLR-4*, and *TNF-α* did not differ significantly between DMSO- and RPMI-treated cells (i.e., the normal condition of RAW264.7 cells), indicating that 0.1% DMSO had no effect on proinflammatory cytokine expression levels. LPS-treated cells were used as a positive control, showing increased expression levels of immune-related genes, such as *IL-1β*, *IL-6*, *COX-2*, *iNOS*, *TLR-4*, and *TNF-α*. Compared with all treatments other than LPS, treatment with 100 µg/mL of DJ-PPD significantly increased the mRNA expression levels of all immune-related genes except *iNOS*. Furthermore, compared with DJ-treated (normal rice without PPD) cells, DJ-PPD, S-PPD, and GE-treated cells exhibited significantly increased expression levels of all tested immune-related genes. This reliably suggests that the PPD-enriched rice (DJ-PPD) might stimulate RAW264.7 cells to produce higher levels of immune-related mRNA compared with other treatments.

### 2.5. PGE_2_ Production

Because PGE_2_ is one of the most important inflammatory mediators, the effects of DJ-PPD in RAW264.7 cells and LPS-stimulated RAW264.7 cells were investigated. According to their NO production, cells treated with the highest test concentrations exhibited the highest immunomodulatory activity through the inhibition of LPS-induced NO production levels in LPS-stimulated RAW264.7 cells and the enhancement of NO production in RAW264.7 cells. Therefore, to evaluate PGE_2_ production, DJ, DJ-PPD, and GE were tested at 100 µg/mL, while S-PPD was tested at 700 pg/mL. In LPS-stimulated RAW264.7 cells, all pretreatments significantly inhibited PGE_2_ production compared with that in DJ-treated cells (Figure 5a). In contrast, treatment with DJ-PPD, S-PPD, and GE markedly enhanced PGE_2_ production compared with DJ (normal rice without PPD) in RAW264.7 cells (Figure 5b). Moreover, DJ-PPD-treated cells exhibited the highest inhibition and enhancement of PGE_2_ production in LPS-stimulated and unstimulated RAW264.7 cells, respectively. It is evident that DJ-PPD, which produces high concentrations of PPD, regulates the production of PGE_2_ depending on the cell condition. DJ-PPD induced the production of PGE_2_ under normal conditions (RAW264.7 cells), whereas it decreased LPS-induced PGE_2_ production in cells under inflammatory conditions.

### 2.6. Phagocytosis Activity

Phagocytosis is a host defense mechanism that reflects a host’s immune status [19]. Compared with the phagocytic activity in RPMI-treated cells, phagocytosis was enhanced in DJ-, DJ-PPD-, S-PPD-, GE-, and LPS-treated cells (Figure 6). The positive control treatment with 1 µg/mL of LPS significantly increased phagocytosis in RAW264.7 cells relative to all other treatments. Notably, treatment with DJ-PPD markedly increased phagocytic activity in RAW264.7 cells relative to treatment with DJ; however, phagocytosis did not differ among RAW264.7 cells treated with DJ-PPD, S-PPD, and GE containing PPD-type sapogenin or saponins.

### 2.7. Effects of Various Treatments on NF-κB and MAPK Pathway Activation

As shown in Figure 7a, LPS treatment (DMSO group) induced NF-κB and MAPK pathway activation through the increased levels of phosphorylated (p-)NF-κB p65, p-p38 MAPK, and p-JNK. However, treatment with DJ, DJ-PPD, S-PPD, and GE significantly downregulated LPS-induced NF-κB p65, p38 MAPK, and p-JNK phosphorylation. Among all treatments, DJ-PPD resulted in the highest inhibition of NF-κB p65 and JNK phosphorylation, whereas the highest inhibition of p38 MAPK phosphorylation was found in both DJ-PPD- and S-PPD-treated cells. In GE-treated cells, p-NF-κB p65, p-p38 MAPK, and p-JNK levels were significantly higher than those in DJ-PPD-treated cells; however, these proteins levels did not differ significantly among S-PPD- and GE-treated cells.

In RAW264.7 cells, the DJ-PPD, S-PPD, and GE treatments significantly increased p-NF-κB p65, p-p38 MAPK, and p-JNK levels compared to DJ (normal rice without PPD) treatment (Figure 7b). Compared with the S-PPD and GE treatments, the DJ-PPD treatment significantly upregulated p-NF-κB p65 and p-JNK levels but did not significantly affect p-p38 MAPK levels. These results further prove that DJ-PPD (PPD-enriched transgenic rice) can activate the NF-κB and MAPKs pathways in RAW 264.7 cells and inactivate these pathways in LPS-stimulated RAW264.7 cells, consistent with the above results.

## 3. Discussion

NF-κB is an inducible transcription factor. Generally, NF-κB and IκB proteins share an interaction and are present in an inactive form in the cytosol [20]. NF-κB activation plays an important role in immune and inflammatory responses [21], including the production of proinflammatory cytokines, chemokines, and inflammatory mediators, such as IL-1, IL-6, IL-12, COX-2, iNOS, and TNF-α, in different types of innate immune cells [21,22]. Similarly, the MAPK pathway is related to the immune response. The activation of p38 MAPK and JNK induces the production of immunomodulatory cytokines, such as IL-1β, IL-6, and TNF-α [23]. Apart from releasing immunomodulatory cytokines, the activation of the MAPK signaling pathway also promotes the production of prostaglandins, which exert immune-related biological effects [24,25]. Conversely, activation of the NF-κB pathway promotes differentiation of Th1 and Th17 cells (i.e., inflammatory T cells) [26,27], and activation of the MAPK pathway plays a critical role in regulating Th1- and Th2-type responses [28]. The results of the current study show that treatment with DJ-PPD, which produces high concentrations of PPD, significantly increased the production of p-NF-κB p65 and p-JNK levels compared to cells treated with GE and S-PPD (Figure 7b and Figure 8, respectively). However, p-p38 MAPK expression did not differ among DJ-PPD-, GE-, and S-PPD-treated cells. Moreover, the increased expression of NF-κB and MAPK pathway activation proteins affected the release of immunomodulatory cytokines. In particular, *IL-1β*, *IL-6*, *COX-2*, and *TNF-α* expression was markedly increased in DJ-PPD-treated cells (Figure 4). In addition, the expression of *TLR-4*, an NF-κB pathway activation mediator [29], was higher in DJ-PPD-treated cells than in GE- and S-PPD-treated cells. *COX-2* expression plays a rate-determining role in PGE_2_ production [25], and TNF-α and IL-6 have been reported as enhancers of macrophage phagocytosis [30,31,32]. The increase in IL-6 and TNF-α promoted phagocytosis in RAW264.7 cells. According to our results, phagocytotic activity was significantly correlated with TNF-α and IL-6 expression levels (Pearson’s correlation coefficient = 0.987 and 0.989, respectively; critical value for Pearson’s r at a degree of freedom = 5 and *p* < 0.01 = 0.874). We also observed a similar trend of PGE_2_ production to the COX-2 expression pattern (Figure 4) in treated RAW264.7 cells (Figure 5b). Phagocytic activity markedly increased in DJ-PPD-, GE-, and S-PPD-treated cells (100 µg/mL or 700 pg/mL) compared with that in DJ (normal rice without PPD)-treated cells, although without significant differences among the three treatment groups (Figure 6).

Inflammation is a defense mechanism used by host bodies to eliminate harmful and foreign stimuli [33]. During infection, the host produces several proinflammatory cytokines that are known to play critical roles in the pathogenesis of diseases [34]. Although cytokines such as IL-6 are essential, their constitutive overproduction is often involved in diseases related to acute and chronic inflammation [35,36]. Therefore, suppressing the excess expression and production of powerful mediators via anti-inflammatory components represents one possible preventive and/or therapeutic strategy. In the present study, we used LPS to mimic an inflammation environment [37,38,39] in RAW264.7 cells; LPS treatment led to the activation of the NF-κB and MAPK pathways via upregulation of *TLR-4*, a key receptor involved in LPS recognition [40]. Treatment with PPD-enriched transgenic rice (DJ-PPD) increased inhibition of p-NF-κB p65, p-p38 MAPK, and p-JNK expression relative to GE at the same concentration (100 μg/mL, Figure 7a). Additionally, treatment with 100 µg/mL of DJ-PPD significantly suppressed p-NF-κB p65 and p-JNK compared with that in S-PPD-treated cells. The downregulation of these proteins further decreased the levels of LPS-induced inflammation biomarkers such as NO production (Table 1), *IL-1β*, *IL-6*, *COX-2*, *iNOS*, *TLR-4*, and *TNF-α* expression (Figure 3), as well as PGE_2_ production (Figure 5a). Compared with RPMI treatment, DJ treatment also significantly reduced the production of LPS-induced inflammatory factors. Thus, some compounds in normal rice and the PPD in transgenic rice (DJ-PPD) may have a synergic effect. Consequently, treatment with DJ-PPD led to increased immune enhancement and anti-inflammatory effects compared with PPD (S-PPD) and GE at the same concentrations.

## 4. Materials and Methods

### 4.1. Materials and Reagents

A SpectraMax^®^ ABS Plus microplate reader was purchased from Molecular Device (San Jose, CA, USA). A CFX Connect real-time PCR system, CFX Maestro software, a ChemiDoc imaging system, and Clarity™ Western ECL substrate were purchased from Bio-Rad (Hercules, CA, USA). An IM-3 series microscope was obtained from Optika (Bergamo, Italy). Filter papers (5 µm) were obtained from Hyundai Micro (Seoul, Republic of Korea). 20s-PPD with a purity of ≥98% HPLC (S-PPD) was obtained from Ambo Institute (Daejeon, Republic of Korea). RPMI-1640 medium and fetal bovine serum (FBS) were acquired from Gibco™ (Thermo Fisher Scientific, Inc., Waltham, MA, USA). Penicillin/streptomycin (P/S) was purchased from Hyclone Laboratories, Inc (Logan, UT, USA). Griess reagent, neutral red, and Bradford reagent were obtained from Sigma-Aldrich (St. Louis, MO, USA). An EZ-CyTox cell viability assay kit was purchased from DoGenBio (Seoul, Republic of Korea). TRI reagent™ was obtained from Invitrogen (Waltham, MA, USA). A Power cDNA synthesis kit and RealMOD™ Green W^2^ 2× qPCR mix were purchased from Intron Biotechnology (Seongnam-Si, Republic of Korea). A PGE_2_ ELISA kit was obtained from Enzo Life Science (Farmingdale, NY, USA). RIPA buffer was obtained from Geneall Biotechnology (Seoul, Republic of Korea), and Protease Inhibitor Cocktail Kit 5 was purchased from Bio-Medical Science (Seoul, Republic of Korea). The primary antibodies for p-NF-κB p65, p-p38 MAPK, and p-JNK were purchased from Cell Signaling (Danvers, MA, USA). GAPDH and m-IgGκ BP-HRP antibodies were obtained from Santa Cruz Biotechnology (Dallas, TX, USA). Goat anti-rabbit IgG(H + L)-HRP was purchased from GenDEPOT (Baker, TX, USA).

### 4.2. Sample Preparation

Rice seed and dried-ginseng root (5 years old) were extracted using a previously described method [16]. The extracts were prepared at concentrations of 10, 25, 50, and 100 mg/mL in DMSO. S-PPD was prepared at concentrations of 70, 175, 350, and 700 ng/mL based on the amount of PPD in DJ-PPD extract (7.28 ± 0.64 µg of PPD/g of rice seed extract) [16].

### 4.3. Cell Culture and Treatment

RAW264.7 macrophage cells (Korean Cell Line Bank; KCLB No. 4071, Seoul, Republic of Korea) were maintained in RPMI-1640 medium (containing 10% FBS and 1% P/S). The cells were kept under incubator-controlled conditions at 37 °C and 5% CO_2_. They were seeded at 1 × 10^5^ cells/well, 5 × 10^5^ cells/well, and 2 × 10^6^ cells/well in 96-, 24-, and 6-well plates, respectively. These plates were incubated in an incubator for 24 h at 37 °C with 5% CO_2_. Cultured media were replaced by extracts at final concentrations of 10, 25, 50, and 100 µg/mL in RPMI-1640 medium. In the S-PPD group, final concentrations of 70, 175, 350, and 700 pg/mL in RPMI-1640 medium were applied to the cells. After 1 h of pretreatment, cells were stimulated with 1 µg/mL of LPS or not stimulated at all, and the plates were incubated for either 6 or 24 h.

### 4.4. Cell Viability and NO Production Assays

The stimulated cells in 96-well plates were used to determine cell viability with an EZ-CyTox cell viability assay kit. A working solution of EZ-CyTox (EZ-CyTox: 1× phosphate-buffered saline (PBS) (1:10)) was added to each well (110 µL/well). The plates were maintained at 37 °C. After 4 h of incubation, 100 µL of the solution in each well was transferred to new plates. The absorbance at 450 nm was measured using a SpectraMax^®^ ABS Plus microplate reader, and cell viability was calculated based on the following formula:(1)$$\mathrm{Cell}\;\mathrm{viability}\;\mathrm{ratio}\;\left(\%\right)=\frac{\mathrm{absorbance}\;\mathrm{at}\;450\;\mathrm{nm}\;\mathrm{of}\;\mathrm{treatment}\;-\;\mathrm{absorbance}\;\mathrm{at}\;450\;\mathrm{nm}\;\mathrm{of}\;\mathrm{blank}\;}{\mathrm{absorbance}\;\mathrm{at}\;450\;\mathrm{nm}\;\mathrm{of}\;\mathrm{control}\;-\;\mathrm{absorbance}\;\mathrm{at}\;450\;\mathrm{nm}\;\mathrm{of}\;\mathrm{blank}}\;\times \;100.$$

NO production was investigated using cultured media and Griess reagent, which were mixed in a ratio of 1:1 (*v*/*v*) and incubated at room temperature for 15 min shielded from the light. The absorbance at 540 nm was measured and quantified by constructing a standard curve of sodium nitrite.

### 4.5. RNA Isolation and cDNA Synthesis

After 6 h of LPS stimulation, cells in a 24-well plate were collected and washed with ice-cold 1× PBS. Cell lysis was performed using TRI reagent™ at room temperature. RNA precipitation was performed by the addition of 500 µL of 100% isopropanol and incubation at 4 °C for 30 min. The RNA pellet was collected via centrifugation at 13,000 rpm and 4 °C for 10 min. The pellet was washed with 70% ethanol three times and dried by opening the tube cap. The pellet was then resuspended in nuclease-free water and RNA-quantified using a SpectraMax^®^ ABS Plus microplate reader. First-strand cDNA was synthesized using 500 ng of total RNA and a Power cDNA synthesis kit according to manufacturer’s instructions.

### 4.6. Expression Analysis of Immune-Related Genes via Real-Time PCR

The transcripts of immune-related genes were evaluated using real-time PCR in a CFX Connect real-time PCR system using RealMOD™ Green W^2^ 2× qPCR mix with specific primers. The sequences of the primers used were as follows: *IL-1β* (NM_008361.4), forward 5′-GGG CCT CAA AGG AAA GAA TC-3′ and reverse 5′-TAC CAG TTG GGG AAC TCT GC-3′; *IL-6* (NM_031168.2), forward 5′-AGT TGC CTT CTT GGG ACT GA-3′ and reverse 5′-CAG AAT TGC CAT TGC ACA AC-3′; *COX-2* (NM_011198.4), forward 5′-AGA AGG AAA TGG CTG CAG AA-3′ and reverse 5′-GCT CGG CTT CCA GTA TTG AG-3′; *iNOS* (BC062378.1), forward 5′-TTC CAG AAT CCC TGG ACA AG-3′ and reverse 5′-TGG TCA AAC TCT TGG GGT TC-3′; *TLR-4* (NM_021297.3), forward 5′-CGC TCT GGC ATC ATC TTC AT-3′ and reverse 5′-GTT GCC GTT TCT TGT TCT TCC-3′; *TNF-α* (D84199.2), forward 5′-ATG AGC ACA GAA AGC ATG ATC-3′ and reverse 5′-TAC AGG CTT GTC ACT CGA ATT-3′; and *β-actin* (NM_007393.5), forward, 5′-CCA CAG CTG AGA GGA AAT C-3′ and reverse 5′-AAG GAA GGC TGG AAA AGA GC-3′. The PCR reaction conditions were as follows: preheating at 95 °C for 10 min; 40 cycles at 95 °C for 20 s, 60 °C for 20 s, and 72 °C for 30 s; and a final extension at 72 °C for 5 min. Gene expression levels (fold changes) were analyzed using CFX Maestro software, with *β-actin* used as a reference gene. The parameters for gene expression analysis were assigned as follows: reference gene = *β-actin*; mode = normalized expression (ΔΔCq); graph data = relative to control; control = media; error bar = standard deviation.

### 4.7. PGE_2_ Production

PGE_2_ production in the cultured media was determined using a PGE_2_ ELISA kit. To avoid false results due to cell confounders, the supernatant was centrifuged at 3000 rpm for 10 min. One hundred microliters from each treatment group was used for PGE_2_ quantification. The experiment was performed in duplicate, and the untreated group was used as a negative control.

### 4.8. Phagocytosis Assay

The effects of DJ, DJ-PPD, S-PPD, and GE treatment on phagocytosis in RAW264.7 cells was investigated using a neutral red uptake method [41] with slight modifications. Briefly, 0.075% of neutral red was added to each well, and cells were incubated for 2 h at room temperature shielded from the light. The excess dye was washed out using 1× PBS (five times). After drying, an IM-3 series microscope was used to capture cell images. Afterward, lysis solution (50% ethanol:glacial acetic acid (1:1)) was added to each well. After 2 h of lysis, the absorbance was measured at 540 nm. The phagocytic activity was then calculated by comparison with the RPMI group.

### 4.9. Western Blot Assay

Protein was extracted from cells in 6-well plates using RIPA buffer supplemented with 1× Protease Inhibitor Cocktail Kit 5 on ice for 30 min and collected via centrifugation at 13,000 rpm and 4 °C for 30 min. The extracted proteins were measured using Bradford reagent and quantified using a bovine serum albumin standard curve. Thirty micrograms of protein from each treatment were separated using 10% SDS-PAGE and transferred to a nitrocellulose membrane. Membranes were incubated with primary antibodies specific to p-NF-κB p-65, p-p38 MAPK, p-JNK, and GAPDH at 4 °C overnight. After washing, the membranes were incubated with the secondary antibodies (goat anti-rabbit IgG(H + L)-HRP (p-NF-κB p-65, p-p38, and p-JNK) or m-IgGκ BP-HRP (GAPDH)) for 1 h at room temperature. Protein signaling was detected using Clarity™ Western ECL substrate (Bio-Rad, Hercules, CA, USA), and the detected signals were imaged and quantified in terms of intensity using a ChemiDoc imaging system (Bio-Rad, Hercules, CA, USA).

### 4.10. Statistical Analysis

Data are shown as means ± standard deviations. Statistix (version 8.1; Statistix, Tallahassee, FL, USA) was used to conduct statistical analyses. Data were analyzed using one-way analysis of variance followed by post hoc Duncan’s multiple range tests. Differences between two groups were assessed using *t*-tests at a significance level of *p* < 0.05.

## 5. Conclusions

In conclusion, our study demonstrates the immuno-boosting effects of PPD-enriched rice (DJ-PPD) in RAW264.7 cells and its anti-inflammatory activity in LPS-stimulated RAW264.7 cells (Figure 8). Compared with the S-PPD and GE groups, treatment with the same concentration of DJ-PPD led to higher anti-inflammatory activities in RAW264.7 cells under LPS-induced inflammation. However, DJ-PPD treatment exhibited similar or higher immune enhancement effects than those exerted by either S-PPD or GE. The results of the present study show that PPD-enriched rice seeds that do not require complex intestinal microbial processes might be considered a potentially useful natural immunomodulatory agent.

**Figure 8 plants-12-00767-f008:**
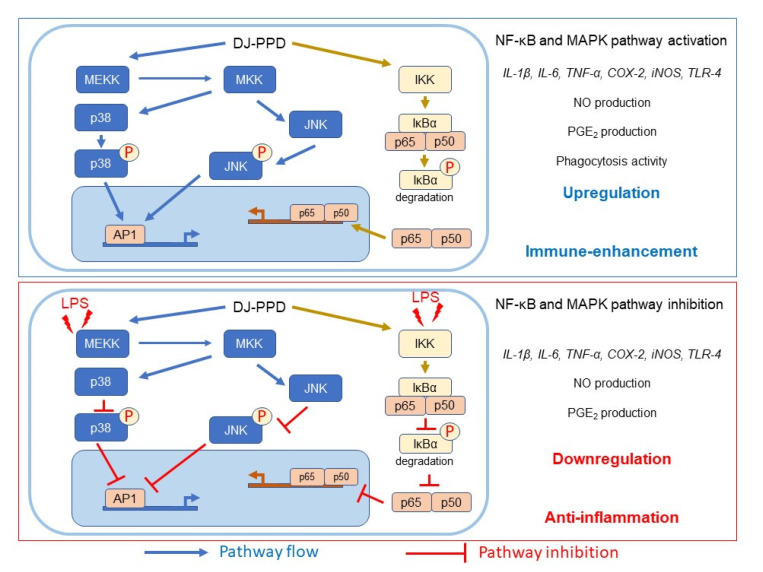
Schematic diagram of the effects of DJ-PPD on NF-κB and MAPK pathway regulation.

## Figures and Tables

**Figure 1 plants-12-00767-f001:**
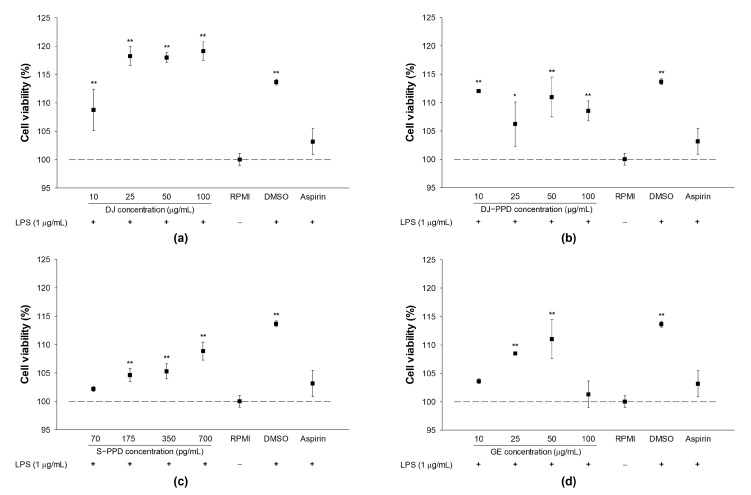
Cell viability in cells pretreated with (**a**) normal rice seed extract (DJ), (**b**) protopanaxadiol-enriched transgenic rice seed extract (DJ-PPD), (**c**) 20s-protopanaxadiol (S-PPD), and (**d**) natural ginseng root extract (GE) followed by LPS stimulation. After the cells were stimulated with or without LPS, they were incubated for an additional 24 h. DMSO and aspirin concentrations were 0.1% and 200 µg/mL, respectively. Data are shown as means ± standard deviation (*n* = 3). Significant differences at *p* < 0.05 (*) and *p* < 0.01 (**) were determined by comparison with the RPMI group.

**Figure 2 plants-12-00767-f002:**
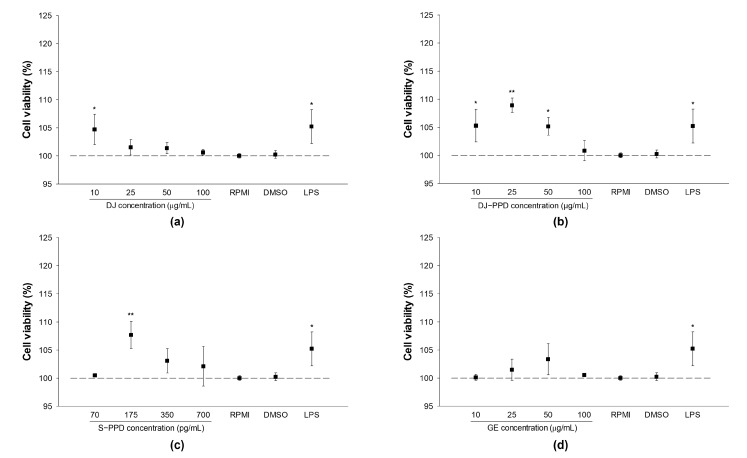
Cell viability in cells pretreated with (**a**) DJ extract, (**b**) DJ-PPD extract, (**c**) S-PPD, and (**d**) GE extract. Treated cells were incubated for 24 h before cell viability measurement. The concentration of DMSO and LPS was 0.1%. LPS was used as a positive at 1 µg/mL. Data are shown as means ± standard deviation (*n* = 3). Significant differences at *p* < 0.05 (*) and *p* < 0.01 (**) were determined by comparison with the RPMI group.

**Figure 3 plants-12-00767-f003:**
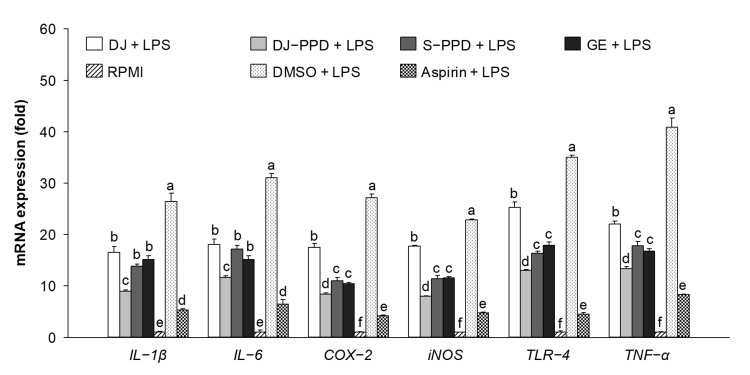
Inflammation-related mRNA suppression in cells pretreated with DJ extract, DJ-PPD extract, S-PPD, and GE extract, together with LPS stimulation. After 6 h of LPS stimulation, cells were collected for RNA extraction. DJ, DJ-PPD, and GE concentrations were 100 µg/mL, whereas that of S-PPD was 700 pg/mL. DMSO and aspirin concentrations were 0.1% and 200 µg/mL, respectively. Data are shown as means ± standard deviations. Different lowercase letters indicate significant differences among treatments at *p* < 0.05.

**Figure 4 plants-12-00767-f004:**
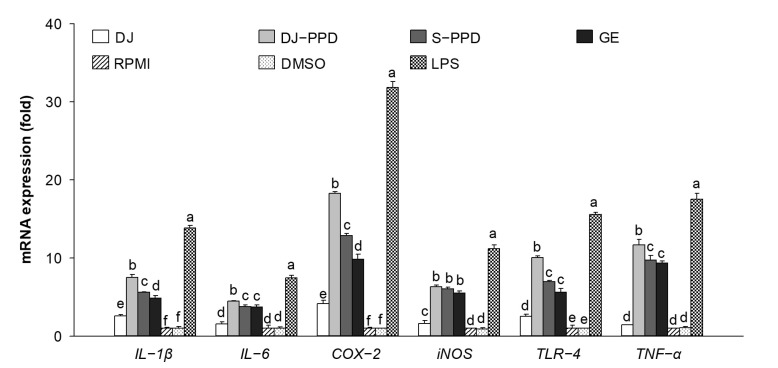
Inflammation-related mRNA upregulation in cells pretreated with DJ extract, DJ-PPD extract, S-PPD, and GE extract. Cells were collected for RNA extraction after 6 h of incubation. DJ, DJ-PPD, and GE concentrations were 100 µg/mL, whereas the S-PPD concentration was 700 pg/mL. DMSO and LPS concentrations were 0.1% and 1 µg/mL, respectively. Data are shown as means ± standard deviations. Different lowercase letters indicate significant differences among treatments at *p* < 0.05.

**Figure 5 plants-12-00767-f005:**
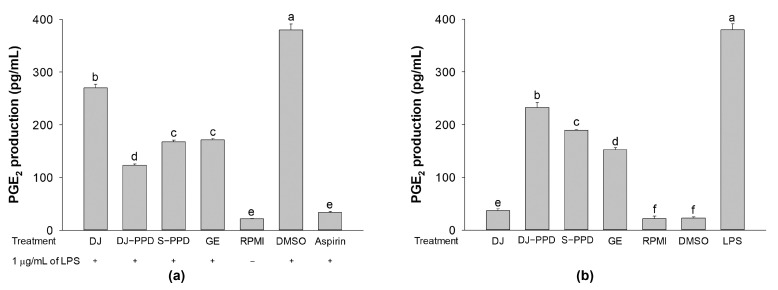
Regulation of PGE_2_ production in cells pretreated with DJ extract, DJ-PPD extract, S-PPD, and GE extract. (**a**) The downregulation effect in LPS-stimulated RAW264.7 cells and (**b**) the upregulation effect in unstimulated RAW264.7 cells. After the cells were stimulated with or without LPS, they were incubated for an additional 24 h. The supernatant from each treatment was collected, and the produced PGE_2_ was measured. DJ, DJ-PPD, and GE concentrations were 100 µg/mL, whereas the S-PPD concentration was 700 pg/mL. DMSO, aspirin, and LPS concentrations were 0.1%, 200 µg/mL, and 1 µg/mL, respectively. Data are shown as means ± standard deviations. Different lowercase letters indicate significant differences among treatments at *p* < 0.05.

**Figure 6 plants-12-00767-f006:**
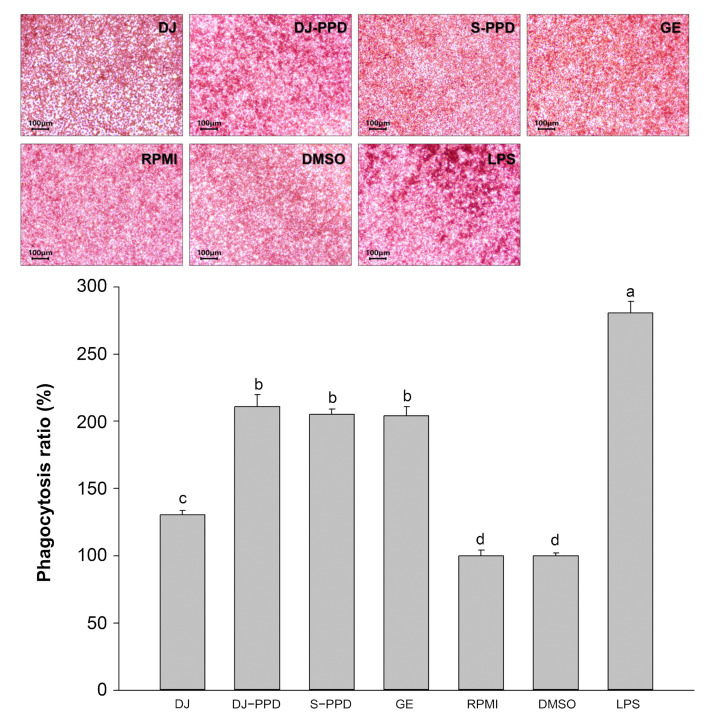
Increasing phagocytic activity in RAW264.7 cells treated with DJ extract, DJ-PPD extract, S-PPD, and GE extract. Cells were incubated for 24 h after pretreatment with each sample. DJ, DJ-PPD, and GE concentrations were 100 µg/mL, whereas the S-PPD concentration was 700 pg/mL. DMSO and LPS concentrations were 0.1% and 1 µg/mL, respectively. Data are shown as means ± standard deviations. Different lowercase letters indicate significant differences among treatments at *p* < 0.05.

**Figure 7 plants-12-00767-f007:**
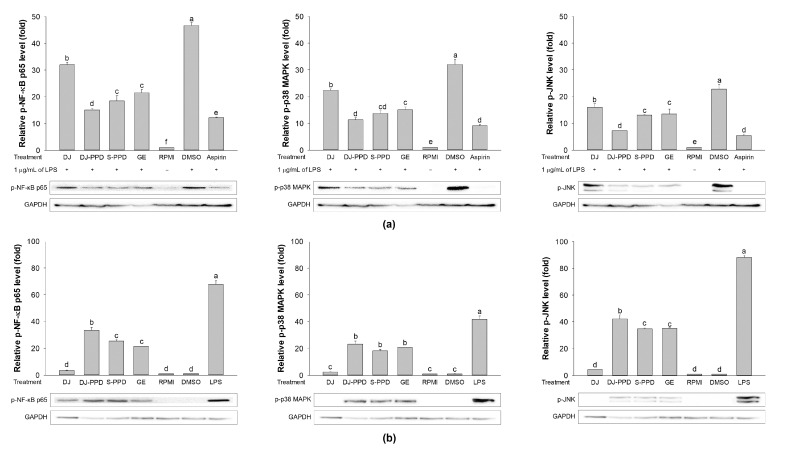
The immune-related pathway activation, (**a**) downregulation in cells pretreated with DJ extract, DJ-PPD extract, S-PPD, and GE extract, together with LPS stimulation, and (**b**) upregulation in cells pretreated with DJ extract, DJ-PPD extract, S-PPD, and GE extract. DJ, DJ-PPD, and GE concentrations were 100 µg/mL, whereas the S-PPD concentration was 700 pg/mL. DMSO and LPS concentrations were 0.1% and 1 µg/mL, respectively. Data are shown as means ± standard deviations. Lowercase letters indicate significant differences among treatments at *p* < 0.05.

**Table 1 plants-12-00767-t001:** Pretreatment of DJ, DJ-PPD, S-PPD, and GE suppressed LPS-induced NO production in RAW264.7 cells.

Treatment	NO Production (µM)			
	10 µg/mL (70 pg/mL of S-PPD)	25 µg/mL (175 pg/mL of S-PPD)	50 µg/mL (350 pg/mL of S-PPD)	100 µg/mL (700 pg/mL of S-PPD)
DJ + LPS	31.03 ± 0.40 ^a,^*	29.21 ± 0.58 ^a,^*	26.36 ± 0.34 ^a,^*	24.25 ± 0.31 ^a,^*
DJ−PPD + LPS	29.47 ± 0.37 ^b,^*	24.80 ± 0.38 ^c,^*	21.89 ± 0.36 ^d,^*	16.81 ± 0.25 ^c,^*
S−PPD + LPS	29.60 ± 0.44 ^b,^*	27.65 ± 0.55 ^b,^*	22.94 ± 0.37 ^c,^*	19.41 ± 0.70 ^b,^*
GE + LPS	29.92 ± 0.15 ^b,^*	26.09 ± 0.43 ^c,^*	24.02 ± 0.07 ^b,^*	20.03 ± 0.83 ^b,^*
RPMI	0.00 ± 0.10 *			
DMSO + LPS	33.53 ± 0.43			
Aspirin + LPS	6.32 ± 0.78 *			

After stimulation with or without LPS, cells were incubated for an additional 24 h. Data are shown as means ± standard deviations (*n* = 3). Different lowercase letters indicate significant differences at *p* < 0.05 among treatments at the same concentration (when; a > b > c > d). Significant differences at *p* < 0.05 (*) were determined by comparison with the DMSO group.

**Table 2 plants-12-00767-t002:** DJ, DJ-PPD, S-PPD, and GE treatments enhanced NO production in RAW264.7 cells.

Treatment	NO Production (µM)			
	10 µg/mL (70 pg/mL of S-PPD)	25 µg/mL (175 pg/mL of S-PPD)	50 µg/mL (350 pg/mL of S-PPD)	100 µg/mL (700 pg/mL of S-PPD)
DJ	0.38 ± 0.40 ^c^	0.56 ± 0.09 ^c^	0.73 ± 0.07 ^b^	1.50 ± 0.06 ^b,^*
DJ−PPD	7.20 ± 0.18 ^a,^*	9.74 ± 0.16 ^a,^*	11.71 ± 0.25 ^a,^*	12.12 ± 0.09 ^a,^*
S−PPD	7.28 ± 0.16 ^a,^*	9.13 ± 0.11 ^b,^*	11.26 ± 0.23 ^a,^*	11.91 ± 0.19 ^a,^*
GE	7.12 ± 0.25 ^a,^*	9.68 ± 0.22 ^a,^*	11.46 ± 0.11 ^a,^*	12.05 ± 0.13 ^a,^*
RPMI	0.00 ± 0.10			
DMSO	0.00 ± 0.05			
LPS	34.35 ± 1.22 *			

Treated cells were incubated for 24 h before cell viability measurement. Data are shown as means ± standard deviations (*n* = 3). Different lowercase letters indicate significant differences at *p* < 0.05 among treatments at the same concentration (when; a > b > c). Significant differences at *p* < 0.05 (*) were determined by comparison with the RPMI group.

## Data Availability

The data is contained within the manuscript and Appendix A.

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
