# Peer review of "Effect of Ginseng Sapogenin Protopanaxadiol-Enriched Rice (DJ-PPD) on Immunomodulation"

_plants, 2023, doi:10.3390/plants12040767_

Round 1

Reviewer 1 Report

   This work reports an interesting research on the immunomodulatory properties of Protopanaxadiol (PPD), a tetracyclic terpene sapogenin found in ginseng, based on enriched transgenic rice seed extracts of it (DJ-PPD), and assayed on model cell cultures, and compared with ginseng-derived or synthetic PPD. In fact, is the forth article of a series in which the research group reports the transgenic production of PPD in rice (Han JI et al, 2019), the fundamental immunomodulatory effects of extracts of it on immuno biomarkers (Monmai C. et al, 2022), and on the suppression of oxidative and melanogenic activities in melan-a cells (Monmai C et al, 2023)

    The subject is attractive, with potential useful applications, and within an apparently well-made work, easy to read and with proper controls, including statistics.  However, authors should more explicitly and better explain the connection and differences among such four works (the previously published and the present one), and particularly between the second and the present one.  Also, revise the sentence of lines 21-23 of the abstract that is not easily understood.

    Besides, an important flaw must be corrected in the present manuscript:  the lack of Figures S1, S2 and S3 among those provided to the reviewers, making difficult the proper evaluation of it.

Reviewer 2 Report

This is an in vitro study demonstrating that Protopanaxadiol (PPD), one of the active compounds in ginseng, is endowed with immunomodulatory and inflammation suppressing activity. The study is well conducted, the results are sound. However, it is unclear whether these in vitro proven activities translate into actual immunity boosting effects in the real world. A certain active compound might increase/decrease a certain mediator to a level, say, 4 times superior/inferior to the basal one, but in real life an at least 40 times change in the level of the mediator might be necessary to make a difference. Therefore, the introduction or discussion sections should contain evidence based information demonstrating that changes of the magnitude achieved in this study are sufficient to exert a therapeutic effect on human (or at least animal) subjects.

The article is not reader friendly.

I could not find an abbreviations list.

The legends of figures and tables are not informative enough - formulations such as " Effects of ... on ..." are unacceptable. The legends of figures and tables should convey enough information so that the readers might understand them without referring to the related text in the manuscript. It should be noted that adding a general statemnt such as "Explantions in the text" is still unacceptable.

Reviewer 3 Report

This manuscript presents a continuation of another previously published work.

The results are consistent, presented and discussed with sufficient depth, contributing to the advancement of knowledge in this specific point.

The work demonstrated the immunoboosting effects of the PPD-enriched rice DJ-PPD in RAW264.7 cells and its anti-inflammatory activity in LPS-stimulated RAW264.7 cells.

According to the immunomodulatory effects of DJ-PPD, S-PPD, and GE in the present study, treatment with DJ-PPD led to higher antiinflammatory effects and similar or higher immune enhancement effects than were exerted by either S-PPD or GE.

The results of the present study showed that PPD-enriched rice seeds that do not require complex intestinal microbial processes might be considered a potentially useful natural immunomodulatory agent.

Minor revisions

Only the completion of the work needs to be improved. It is presented as a compilation of results, and can be presented as conclusions and not results, increasing the attractiveness of the work to the reader.

Round 2

Reviewer 1 Report

   As already advanced in my previous review, this work reports an interesting research on an attractive subject, with potential useful applications, and within a well-made work, easy to read and with proper controls, including statistics.  Besides, in the revised version authors made significant efforts to improve the manuscript in the points indicated by the reviewer, and justify them. Now the manuscript is more complete, in this reviewer's opinion.

    Given that the changes are substantial and have been made in important sections of the manuscript (i.e. Abstract, Introduction, Results and Conclusions), it would be wise to revise the English there.  Below two examples of sites of the text that probably would benefit from it. 

-i.e. at line 23, when saying:  "... higher immune-boosting effects (depend on inflammatory biomarkers)  ..." , would the term "depend" be better substituted by "depending or dependent"?

-i.e. at line 76, when saying: "... activities in melan-a cells with the highest powerful among all   ..." , would the term "powerful" be better substituted by "power" or equivalent?